

# ANXA2 promotes osteogenic differentiation and inhibits cellular senescence of periodontal ligament cells (PDLCs) in high glucose conditions

Yanlin Huang[1,2], Jiaye Wang[1,2], Chunhui Jiang[1,2], Minghe Zheng[3], Mingfang Han[1,2], Qian Fang[1,2], Yizhao Liu[1,2], Ru Li[1,2], Liangjun Zhong[1,2] and Zehui Li[1,2]

[1] Hangzhou Normal University, Zhejiang, China
[2] Department of Stomatology, The Affiliated Hospital of Hangzhou Normal University, Zhejiang, China
[3] Department of Stomatology, No.904 Hospital of the Joint Logistics Support Force of the Chinese People's Liberation Army, Jiangsu Province, Wuxi, China

Corresponding author
Zehui Li, 20221307@hznu.edu.cn

## ABSTRACT

**Background**. Periodontal ligament cells (PDLCs) are a major component of the periodontal ligament and have an important role in the regeneration of periodontal tissue and maintenance of homeostasis. High glucose can affect the activity and function of PDLCs in a variety of ways; therefore, it is particularly important to find ways to alleviate the effects of high glucose on PDLCs. Annexin A2 (ANXA2) is a calcium- and phospholipid-binding protein involved in a variety of cellular functions and processes, including cellular cytokinesis, cytophagy, migration, and proliferation.

**Aim**. The aim of this study was to exploring whether ANXA2 attenuates the deleterious effects of high glucose on PDLCs and promotes osteogenic differentiation capacity.

**Methods and results**. Osteogenic differentiation potential, cellular senescence, oxidative stress, and cellular autophagy were detected. Culturing PDLCs with medium containing different glucose concentrations (CTRL, 8 mM, 10 mM, 25 mM, and 40 mM) revealed that high glucose decreased the protein expression of ANXA2 ($p < 0.0001$). In addition, high glucose decreased the osteogenic differentiation potential of PDLCs as evidenced by decreased calcium deposition ($p = 0.0003$), lowered ALP activity ($p = 0.0010$), and a decline in the expression of osteogenesis-related genes ($p = 0.0008$). Moreover, β-Galactosidase staining and expression of p16, p21 and p53 genes showed that it increased cellular senescence in PDLCs ($p < 0.0001$). Meanwhile high glucose increased oxidative stress in PDLCs as shown by ROS ($p < 0.0001$). However, these damages caused by high glucose were inhibited after the addition of 1 μM recombinant ANXA2 (rANXA2), and we found that rANXA2 enhanced autophagy in PDLCs under high glucose conditions.

**Conclusions and discussion**. Therefore, our present study demonstrates that alterations in ANXA2 under high glucose conditions may be a factor in the decreased osteogenic differentiation potential of PDLCs. Meanwhile, ANXA2 is associated with autophagy, oxidative stress, and cellular senescence under high glucose conditions.

## INTRODUCTION

Periodontitis is a chronic multifactorial inflammatory disease which leads to continued destruction of periodontal tissues and eventual tooth loss if not treated. It affects chewing function, aesthetics, and even general health (*Hajishengallis, 2022*), which seriously affects people's quality of life. Severe periodontitis is the sixth most prevalent disease in the world (*Sanz et al., 2020*), and afflicts about 10.8% of adult population (*Frencken et al., 2017*). Periodontitis is closely related to general health, in which Diabetes Mellitus (DM), the chronic metabolic disease, is one of the major risk factors for periodontitis (*Nibali et al., 2022*), and they are more likely to develop severe periodontitis when their diabetes is not controlled (*Kocher et al., 2018*). For better treatment of periodontitis in diabetic patients, a large number of researchers have searched for a causal relationship between periodontitis and diabetes since 1960.

Periodontal ligament cells (PDLCs) are the major component of the periodontium, which includes fibroblasts, osteoblasts, osteoclasts, macrophages, undifferentiated mesenchymal cells, and many other components. Numerous studies (*Seo et al., 2004*; *Shi et al., 2005*; *Kawanabe et al., 2010*; *Ng et al., 2015*) have shown that periodontal ligament stem cells (PDLSC) can differentiate into various types of cells, including osteoblasts, chondrocytes, neuronal cells and other. After transplantation of PDLSC into periodontal defect areas, there were bone and periodontal membrane-like structures formed, suggesting that they can be used for periodontal tissue regeneration (*Liu et al., 2008*; *Liu et al., 2013*, p. 1; *Iwasaki et al., 2019*). In addition, it has been shown that PDLC and PDLSC are similar in that both have high proliferative capacity and multipotent differentiation abilities (*Feng et al., 2010*). But PDLCs is the main target of inflammatory attack in periodontitis, which is closely related to periodontal tissue health. The diabetics are exposure to elevated blood glucose, and periodontal cells can be damaged through multiple pathways under high glucose conditions (*Kim et al., 2006*). It has been shown that high glucose suppresses the growth and differentiation of periodontal cells (*Kato et al., 2016*), leading to reduced osteogenic capacity. Therefore, it is particularly crucial to find ways to minimize the destruction of PDLCs by high glucose.

Annexin A2 (ANXA2), a 36KD calcium-dependent phospholipid-binding protein, were reported expressed in endothelial cells, monocytes, cancer cells and osteoblasts (*Luo & Hajjar, 2013*). It is extensively involved in membrane repair, exocytosis, endocytosis and voltage-dependent calcium channels (*Gillette & Nielsen-Preiss, 2004*). Its role in fibrinolysis, inflammation regulation and immune system activation, as well as tissue damage and repair has been widely reported (*Lim & Hajjar, 2021*). In recent years, the involvement of annexin A2 in osteoblastic mineralization had been detected. ANXA2 overexpression has been found to result in enhanced mineralization and increased ALP activity (*Gillette & Nielsen-Preiss, 2004*). It has been shown that ANXA2 expression is decreased in patients with type 2 diabetes compared to healthy controls (*Galazis et al., 2013*). In addition, it has been suggested that ANXA2 is an early glycosylation product in experimental diabetes (*Ghitescu, Gugliucci & Dumas, 2001*), and sustained hyperglycemia for 7 days reduced ANXA2 expression in human brain microvascular endothelial cells reduced, and it caused to dysfunction

(*Dai et al., 2013*). Therefore, it seems more meaningful to explore whether ANXA2 can improve the osteogenic differentiation ability of PDLCs in a high glucose environment and reduce the lesions of high glucose on PDLCs.

Oxidative stress can cause periodontal tissue damage and decreased osteogenic potential (*Zheng et al., 2019*). Interestingly, ANXA2 has recently been found to be closely associated with oxidative stress. It has been reported that Annexin A2 regulated ROS levels in septic mice through the IL-17 signaling pathway thereby affecting the inflammatory response (*Tian et al., 2024*). However, it has been shown that extracellular vesicles secreted by cells exposed to ROS expressed more ANXA2 to better resist oxidative stress (*Grindheim & Vedeler, 2016*). Therefore, it is particularly valuable to explore the relationship between ANXA2 and oxidative stress in PDLCs exposed to high glucose environments.

In this study, our aims are: (1) to find out the effect of high glucose on ANXA2 expression; (2) to investigate the influence of high glucose on the osteogenic differentiation of PDLCs and whether this process is regulated by ANXA2; (3) to explore the relationship between ANXA2 and oxidative stress in PDLCs exposed to high glucose environments.

## MATERIAL AND METHODS

### Isolation and culture of PDLCs

Thirty healthy (without periodontitis or caries) third molar teeth or premolar teeth that required extraction for orthodontic reasons were obtained from volunteers (10–25 years) as samples. Written consent form was obtained and the study was approved by the Ethics Committee of the Affiliated Hospital of Hangzhou Normal University (2023(E2)-KS-034).

The PDLCs isolation protocol was as follows: the periodontal ligament tissue was cautiously scraped from the middle third of the root and digested with 2 mg/ml of type I collagenase (Biosharp, Heifi, China) at 37 °C for 30 min. Then, the cells were carefully seeded Dulbecco's modified Eagle's medium (DMEM; Cellmax, China), 10% fetal bovine serum (FBS; Sigma-Aldrich, USA), 100 U/mL penicillin, and 100 µg/mL streptomycin (Solarbio, Beijing, China). PDLCs were incubated at 37 °C in a humidified condition containing 5% $CO_2$, and the medium was changed every three days. When the cells reached 80% confluence, they were digested with 0.25% trypsin-EDTA (HyCyte, Suzhou, China). and sub-cultured. Cells of P3-P6 will be taken for this study.

### Cell viability assay

To explore the effects of glucose and rANXA2 on cell viability in PDLCs, we used the cell counting kit-8 (CCK-8; Solarbio, China) according to the manufacturer' s protocol. PDLCs were placed in 96-well plates (5,000 cells per well) and cultured in the condition with 5% CO2, 37 °C for 24 h. Then, PDLCs were incubated with medium containing different concentrations of glucose (Solarbio) (CTRL, 8 mM, 10 mM, 25 mM, 40 mM, 55 mM, and 70 mM), with or without recombinant ANXA2 (rANXA2) (MedChemExpress (MCE), Monmouth Junction, NJ, USA) (0, 0.01 µM, 0.05 µM, 0.10 µM, 0.50 µM, 1.00 µM, and 2.00 µM) for 24 h and 48 h. After washed with PBS, the PDLCs were cultured in 100 µl DMEM containing 10 µl of CCK-8 solution for 2 h. The absorbance of each well was evaluated by a multi-mode reader (Biotek) at 450 nm.

## Presentation of experimental groups and drug concentrations

In the experiment to detect change of protein expression of ANXA2 in PDLCs under high glucose environment, we set five groups: CTRL, 8 mM, 10 mM, 25 mM, and 40 mM (in 6-well plates, $1 \times 10^5$ per well) to observe the lowest glucose concentration that caused the change of ANXA2 protein expression. In the experiments to detect the cell toxicity of r ANXA2, referring to previous studies[21,27], we set up seven groups with concentrations of 0, 0.01 µM, 0.05 µM, 0.10 µM, 0.50 µM, 1.00 µM, and 2.00 µM of rANXA2 solubilized in the growth medium(in 96-well plates, $0.5 \times 10^4$ per well). To figure out the optimal concentration of r ANXA2 to be administered, we dissolved 0, 0.10 µM, 0.50 µM, 1.00 µM, and 2.00 µM of rANXA2 in a high-glucose medium (40 mM) (in 96-well plates, $0.5 \times 10^4$ per well) to test the concentration of rANXA2, and finally determined 1 µM rANXA2 to be used in the later experiments. We set up CTRL (control group), ODM (osteogenic medium), ODM +10 mM, ODM +25 mM, ODM+40 mM, ODM +25 mM+A, ODM+40 Mm+A for the experiments of alizarin red staining (in 6-well plates, $5 \times 10^4$ per well), ALP activity detection and the qPCR of osteogenesis-related genes (no 10 mM) (in 6-well plates, $1 \times 10^5$ per well). CTRL, 10 Mm, 25 mM, 25 mM+A, 40 mM, 40 mM+A, and 40 mM+A were used for cellular senescence related assays (in 6-well plates, $1 \times 10^5$ per well). Subsequently, in the Western blot analysis of γ-HXA2 (in 6-well plates, $1 \times 10^5$ per well), the detection of ROS (in 24-well plates, $1 \times 10^6$ per well), MDA (in 6-well plates, $5 \times 10^5$ per well), and autophagy-related proteins (in 6-well plates, $3 \times 10^5$ per well), we used CTRL, 25 mM, 25 mM+A, 40 mM, 40 mM+A.

## Western Blot Analysis

To assess the effect of glucose on the expression of ANXA2 ($1 \times 10^5$ per well ,Day7 and Day14), γ-H2AX ($1 \times 10^5$ per well ,Day 3) and autophagy-related proteins($3 \times 10^5$ per well ,4h), PDLCs were cultured in 6-well plates for a period of time and western Blot Analysis was performed. Total protein of PDLCs were isolated by RIPA buffer (EpiZyme, Beijing, China). Then the concentrations of protein were evaluated by BCA Kit (EpiZyme China). 10 µg of each protein sample was loaded on SDS-PAGE gel and transferred to PVDF membranes. The membranes were blocked with BSA (EpiZyme) for 15 min and then were submerged in the primary antibody at 4 °C overnight. After washed by TBST, the membranes were incubated with secondary antibodies at room temperature for 2 h. The antibodies were used as follows: ANXA2 (ab178677; 1:1000 diluted; Abcam, Cambridge, UK), γ-H2AX (2577; 1:1000 diluted; CST, USA), p62 (ab109012; 1:10000 diluted; Abcam), LC3-B (ab192890;1:2000diluted; Abcam, USA), β-actin (380624; 1:5,000 diluted; Zenbio, Chengdu, China), goat anti-rabbit IgG (511203; 1:5000 diluted; Zenbio, China). The bands were detected using the electrochemiluminescence plus reagent (Bio-Rad, Hercules, CA, USA). The relative intensity of these bands was calculated by ImageJ software.

## Alkaline Phosphatase (ALP) activity analysis

To examine the degree of PDLCs osteogenic differentiation, the ALP activity assay was performed when cultured in 6-well plates ($1 \times 10^5$ per well) containing osteogenic differentiation medium (DMEM containing 5% FBS, 50 µg/mL ascorbic acid, 1 µM

dexamethasone, and 3 mM β-glycerophosphate) at day 7, with an ALP kit (Beyotime, China). According to manufacturer's protocol, cells were lysed with RIPA lysis buffer (EpiZyme), and then centrifuged for 10 min at $12,000\times$ g. Collected the supernatant to evaluate the ALP activity by the ALP kit, and the protein concentration by BCA kit (EpiZyme), respectively. The ALP absorbance of each well was evaluated by a multi-mode reader (Biotek) at 405 nm, and BCA at 562 nm. The final ALP activity was normalized to total intercellular protein content.

## Alizarin Red S staining

To assess the degree of PDLCs osteogenic differentiation, PDLCs were placed in 6-well plates ($5 \times 10^4$ per well) for 24 h. PDLCs were then incubated in osteogenic medium at different glucose concentrations with or without rANXA2. 21 days later, alizarin red staining was performed. After washing carefully with PBS (Cellmax, Beijing, China) for 3 times, it was fixed with 4% PFA (Solarbio), and then 1 ml of alizarin red staining solution (HyCyte) was added to each well. The wells were rinsed again with PBS for 3 times, and the calcified nodules were observed under an inverted microscope (Nikon, Tokyo, Japan).

## Senescence-Associated β-Galactosidase (SA-β-Gal) Staining

SA-β-Gal Staining was performed to evaluate the cell senescence of PDLCs induced by high glucose. Seeded into 6-well plates with PDLCs ($1 \times 10^5$ per well) for 24 h, then incubated with medium containing different dose of glucose with or without rANXA2 for 3 days. The SA-β-gal activity of PDLCs were examined according to the manufacturer's protocol (Solarbio). Remove the medium, washed the plates 2 times with PBS and fixed in 4% PFA for 5 min at room temperature. Then removed PFA and added SA-beta-Gal staining solution 2ml per well, staining at 37 °C for 24 h. The senescent PDLCs were stained blue, which can be observed by inverted microscope. Randomly selected five images and the relative intensity of SA-β-gal staining was performed by an Image J Analyze.

## Observation of intracellular ROS

ROS Assay Kit (Solarbio) were used to detected the intracellular ROS. PDLCs were subjected to cell crawls in 24-well plates ($1 \times 10^6$ per well). 24 h later, PDLCs were cultured in medium containing different doses of glucose with or without rANXA2 for 8 h. Then cells were incubated with 200 μl of 0.1% DHE diluted in DMEM at 37 °C for 20 min. After washed 3 times with PBS, the intracellular ROS were observed with confocal microscope (OLYMPUS, Tokyo, Japan). Randomly selected five images were calculated by an Image J Analyze.

## Activity of malonaldehyde (MDA)

Activity of malonaldehyde (MDA) is used to assess oxidative damage. PDLCs were cultured in medium containing different doses of glucose with or without rANXA2 for 3 days in 6-well plates ($5 \times 10^5$ per well). The MDA was examined according to the manufacturer's protocol of the MDA Assay Kit (Beyotime). Cells were lysed with RIPA lysis buffer (EpiZyme), and then centrifuged for 10 min at $12,000\times$ g. Collected the supernatant to evaluate the MDA activity and the protein concentration, respectively. The MDA

**Table 1  Primer.**

| Targe gene | | Nucleotide sequence |
|---|---|---|
| RUNX2 | Forward | GGACGAGGCAAGAGTTTCAC |
| | Reverse | GAGGCGGTCAGAGAACAAAC |
| Osterix | Forward | ACTGTCTGCCCAGTGTCTAC |
| | Reverse | CCCACCATGGAGTAGGAGTG |
| BGLAP | Forward | CCCTCACACTCCTCGCCCTATTG |
| | Reverse | CTGGGTCTCTTCACTACCTCGCTG |
| p16 | Forward | GGGTCGGGTAGAGGAGGTG |
| | Reverse | CCATCATCATGACCTGGATCGG |
| p21 | Forward | TCACTGTCTTGTACCCTTGTGCCTC |
| | Reverse | GGCGTTTGGAGTGGTAGAAATCTGTC |
| p53 | Forward | GTCCAGATGAAGCTCCCAGA |
| | Reverse | CAAGAAGCCCAGACGGAAAC |
| GAPDH | Forward | GGACTCATGACCACAGTCCA |
| | Reverse | TCAGCTCAGGGATGACCTTG |

absorbance of each well was evaluated by a multi-mode reader (Biotek) at 532 nm, and BCA at 562 nm. The final MDA activity was normalized to total intercellular protein content.

## Quantitative real-time PCR (qRT-PCR)

qRT-PCR was used to assess the effect of high glucose on the expression of genes related to osteogenesis (PDLCs cultured in 6-well plates, $1 \times 10^5$ per well, containing osteogenic differentiation medium at day 7 and 14) and cellular senescence (PDLCs cultured in 6-well plates, $5 \times 10^5$ per well, in medium at day 3). PDLCs were harvested in TRIzol (Accurate Biology). Total RNA purification was performed by SteadyPure RNA Kit (Accurate Biology) following the manufacturer's protocols. Then 1 µg of total RNA was obtained as the template to synthesize cDNA using the 5× PrimeScript RT Master Mix (Accurate Biology) according to the manufacturer's protocol. Next, 2 µl cDNA in a 20 µl of reaction volume were employed for real-time PCR with SYBR Green Pro Taq HS qPCR Kit (Accurate Biology). Final expression levels of target genes (Table 1) were normalized to GAPDH, the endogenous housekeeping gene. Data were carried out with the $2^{-\triangle\triangle Ct}$ method.

## Statistical analysis

All data were analyzed by GraphPad Prism software (version 6.0, USA) and expressed as the mean ± standard error of the mean (SEM) from three separate experiments. We used Shapiro–Wilk test and Kolmogorov–Smirnov test to check the distribution of data. Deviations were analyzed by one-way analysis of variance (ANOVA). All data were considered statistically significant when $P < 0.05$.

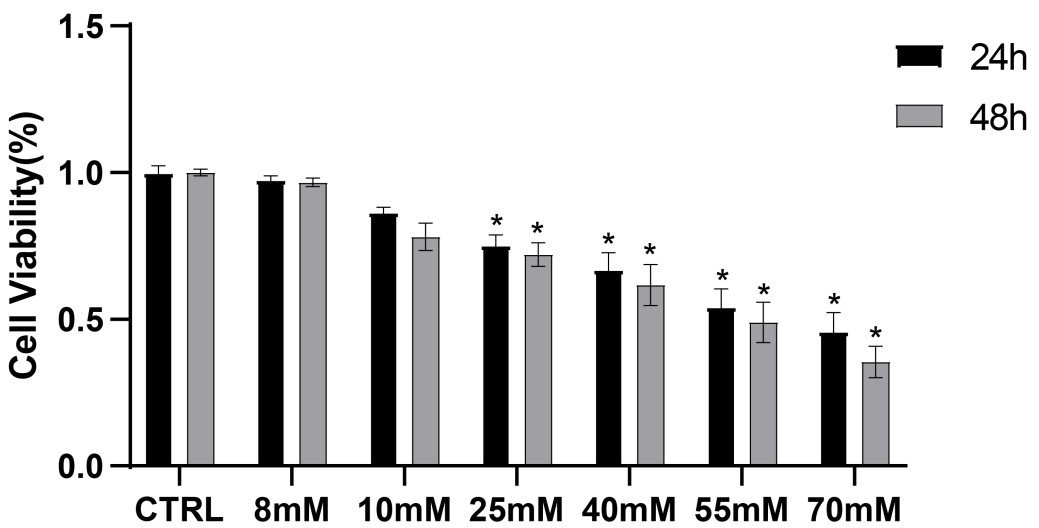

**Figure 1** **CCK8 assay of PDLCs cultured in high glucose.** All data are presented as the means ± SEM from three independent experiments. $*P < 0.05$, compared to CTRL.

## RESULT

### The viability of PDLCs was attenuated in high glucose conditions

To preliminarily investigate the toxic effects of high glucose on PDLCs, PDLCs were treated with medium containing different concentrations of glucose (CTRL, 8 mM, 10 mM, 25 mM, 40 mM, 55 mM, 70 mM) for 24 h and 48 h referring to previous experiments (*Zhang et al., 2019*; *Zheng et al., 2019*; *Mecchia et al., 2022*) . The CCK8 results showed that high glucose began to influence cell viability at 25 mM (Fig. 1).

### ANXA2 expression was decreased in PDLCs under high-glucose condition

To detect whether changes in the expression of ANXA2 in PDLCs were affected by high glucose and find out the lowest glucose concentration caused this change. PDLCs were treating with medium containing different concentrations of glucose (CTRL, 8 mM, 10 mM, 2 5mM, 40 mM) for 7 and 14 days. Then the expression of ANXA2 in the cells was detected by Western blot analysis. The results indicated that at day 7, the dose of 40 mM resulted in a significant decline in the expression of ANXA2 in PDLCs; at day 14, both the doses of 25 mM and 40 mM significantly reduced the expression of ANXA2 compared with CTRL (Fig. 2). In summary, the protein expression of ANXA2 was suppressed when PDLCs were in high glucose condition.

### rANXA2 was not toxic and was able to resist the impact of high glucose on the cellular activity of PDLCs

To test the toxicity of rANXA2 to PDLCs, the PDLCs were cultured in medium containing different concentrations of rANXA2 (0–2 μM) for 24 h and 48 h, with reference to previous studies (*Dai et al., 2013*; *Klabklai et al., 2022*). The 2 μM rANXA2 had no toxicity toward

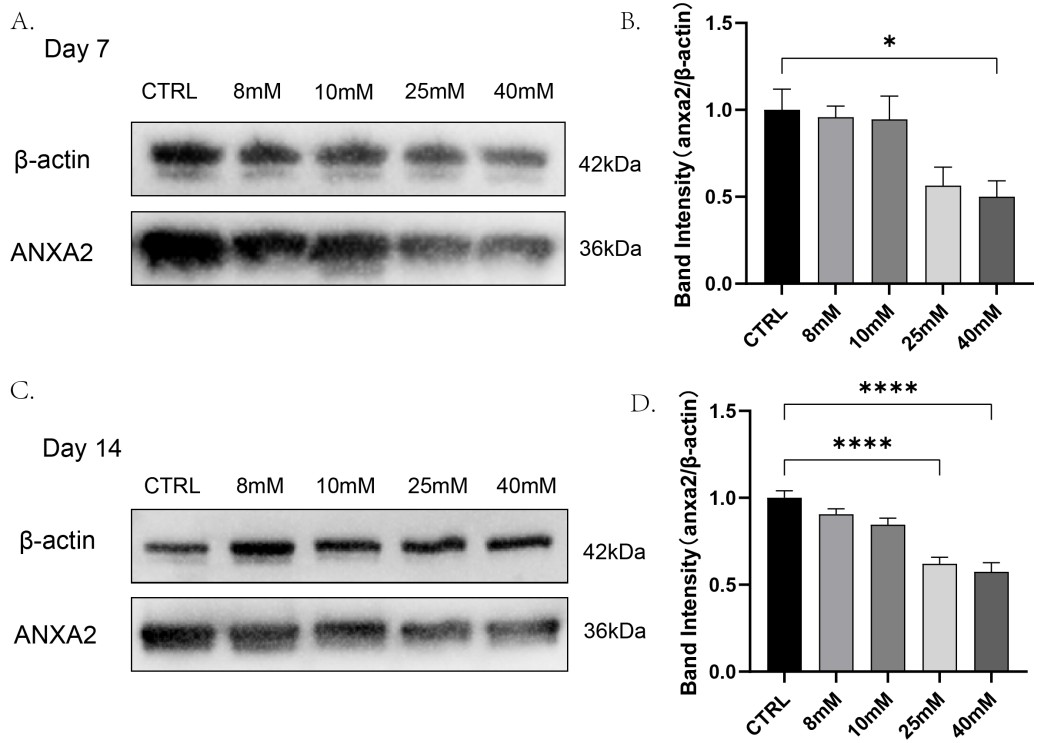

**Figure 2** **The expression of ANXA2 at protein level in PDLCs treated with HG was detected by western blotting assay.** All data are presented as the means $\pm$ SEM from three independent experiments. $^*p < 0.05$, $^{**}p < 0.01$, $^{***}p < 0.001$.

PDLCs (Fig. 3A). Then, to explore the effect of rANXA2 on PDLCs in high-glucose condition, the PDLCs were cultured in medium with 40 mM glucose containing 0–2 μM rANXA2. The result of CCK8 revealed that high glucose (40G) inhibited the PDLCs viability significantly, which can be obstructed by the application of rANXA2 in a dose-dependent manner. The concentrations of 1 μM and 2 μM rANXA2 had the same effect (Fig. 3B). Therefore, 1 μM rANXA2 was used for subsequent experiments.

## High glucose inhibits the osteogenic differentiation of PDLCs, which can be attenuated by administration of rANXA2

In order to explore whether high affected the osteogenic potential of PDLCs and whether this process was regulated by ANXA2, PDLCs were culture in CTRL (control group), ODM (osteogenic medium), ODM +10 mM ODM +25 mM, ODM+40 mM, ODM +25 mM+A, ODM+40 Mm+A for 21 days. The activity of ALP was evaluated on day 7. High glucose inhibited ALP activity in a concentration-dependent manner and the application of rANXA2 at 25G improved ALP activity significantly (Fig. 4D). Then, the mRNA expression levels of RUNX2, Osterix, and BGLAP were assessed at day 7 and day 14 by RT-qPCR (Figs. 4A–4C). Furthermore, matrix mineralization stained by ARS at day 21 can reveal the high glucose inhibited the osteogenic differentiation of PDLCs, which can be attenuated by infusion of rANXA2. The accumulation of ARS in ODM+25 mM, ODM+40 mM, except

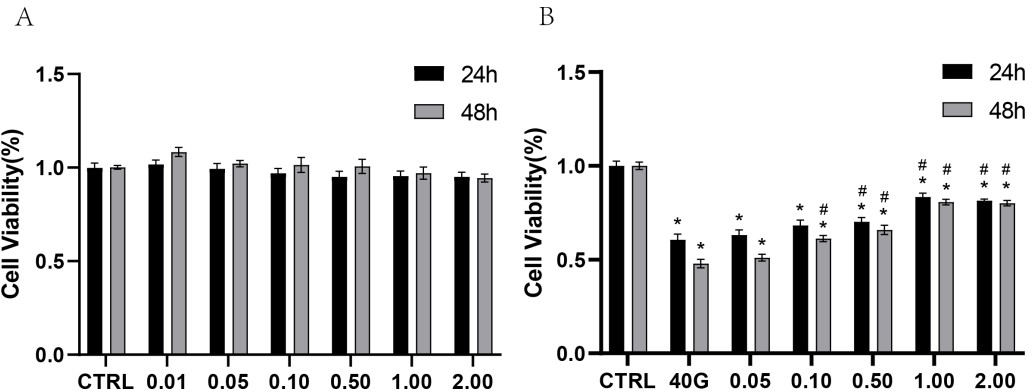

**Figure 3** **CCK8 assay of PDLCs cultured in medium containing different concentrations of rANXA2.** (A) CCK8 assay of PDLCs cultured in medium containing different concentrations of rANXA2. (B) The effect of different concentrations of rANXA2 on viability of PDLCs under 40 mM-glucose after 24 h and 48 h. All data are presented as the means ± SEM from three independent experiments. *$P < 0.05$, compared to CTRL, #$P < 0.05$, compared to 40 mM.

for ODM+10 mM, was less than those in the ODM (Fig. 5), and the application of rANXA2 significantly increased the ARS accumulation of PDLCs differentiated in high-glucose conditions, which examined by a macroscopic and microscopic examination, as well as the semi-quantitative analysis of the ARS staining. The above data indicate that the decreased osteogenic differentiation capacity of PDLCs in a high glucose environment may be due to the reduced expression of ANXA2, which can be attenuated by the administration of rANXA2.

## High glucose induces cellular senescence of PDLCs, which can be inhibited by the application of rANXA2

To find out the effects of high glucose on cellular senescence of PDLCs during osteogenic differentiation and whether ANXA2 was involved in this process, PDLCs were cultured in CTRL, 10 Mm, 25 mM, 25 mM+A, 40 mM, 40 mM+A, and 40 mM+A for 3 days. The result of the senescence-associated beta-galactosidase (SA-β-gal) revealed that cellular senescence was more significant in 25 mM and 40 mM compared to CTRL, and the application of rANXA2 could significantly reduce the production of the senescent cells under 40 mM. And 10 mM group did not yet cause cellular senescence (Figs. 6A–6B). Then, the mRNA levels of the senescence-relative genes revealed that at 40 mM, the expression of *p53*, *p21* and *p16* were significantly increased compared to CTRL. Furthermore, the application of rANXA2 to 40 mM could decrease the expression of *p53*, *p21*, *p16* (Figs. 6C–6E). From the above results, rANXA2 may be able to slow down high glucose-induced cellular senescence of PDLCs *in vitro*.

## rANXA2 reduce oxidative damage of PDLCs induced by high glucose

To explore whether ANXA2 was involved in oxidative damage caused by high glucose in PDLCs, PDLCs were treated with CTRL, 25 mM, 25 mM+A, 40 mM, 40 mM+A. 8 h later, the ROS generation was evaluated by DHE staining (Figs. 7A–7B.) The results indicated

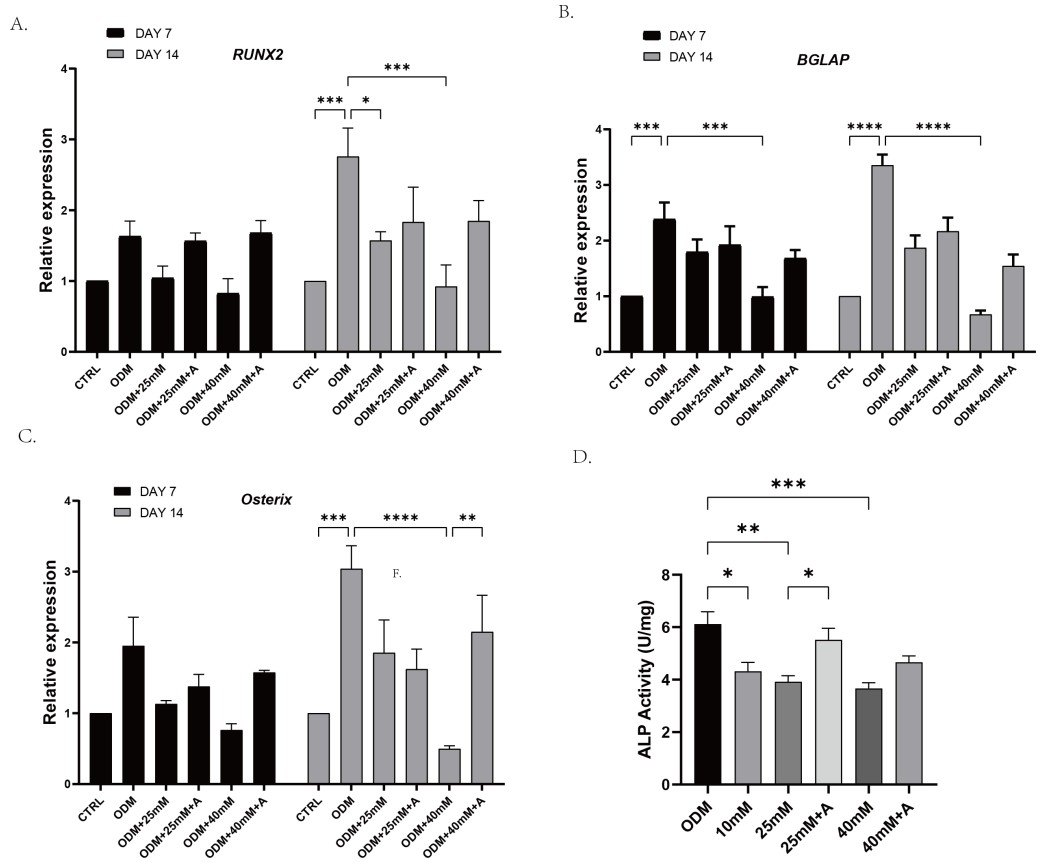

**Figure 4   The effect of high glucose and ANXA2 on osteoblastic differentiation of PDLCs.** The PDLCs were culture in growth medium (control, CTRL), ODM (5.5 mM glucose), ODM with 10mM glucose (10 mM), ODM with 25 mM glucose (25 mM), ODM with 25 mM glucose +1 μM rANXA2 (25 mM +A), ODM with 40 mM glucose (40 mM), ODM with 40 mM glucose +1 μM rANXA2 (40 mM +A). (A–C) The mRNAs of RUNX2, Osterix, and BGLAP were evaluated by qRT-PCR at day 7 and day 14. (D) The ALP activity of PDLCs at day 7. All data are presented as the means ± SEM from three independent experiments. $*p < 0.05$, $**p < 0.01$, $***p < 0.001$.

that high glucose contributed to a significant increase in the production of ROS, which can be inhibited by the addition of rANXA2. Then, 72 h later, we tested for MDA, a biomarker of oxidative damage of lipid. High glucose (40 mM) can cause an elevation of MDA in PDLCs. However, the current results did not provide statistical evidence that rANXA2 can inhibit this process (Fig. 7C). Additionally, the current findings cannot statistically indicate the protein expression of γ-HXA2, a biomarker of DNA damage, is elevated in high glucose condition and is inhibited by the addition of rANXA2, although there was a slight trend as seen in the figure (Figs. 7D–7E). From the above results, rANXA2 may reduce oxidative damage of PDLCs induced by high glucose.

### rANXA2 enhances autophagy in PDLCs under high glucose conditions

To find the relationship of rANXA2 and autophagy in PDLCs under high glucose conditions, the PDLCs were treated with CTRL, 25 mM, 25 mM+A, 40 mM, 40 mM+A

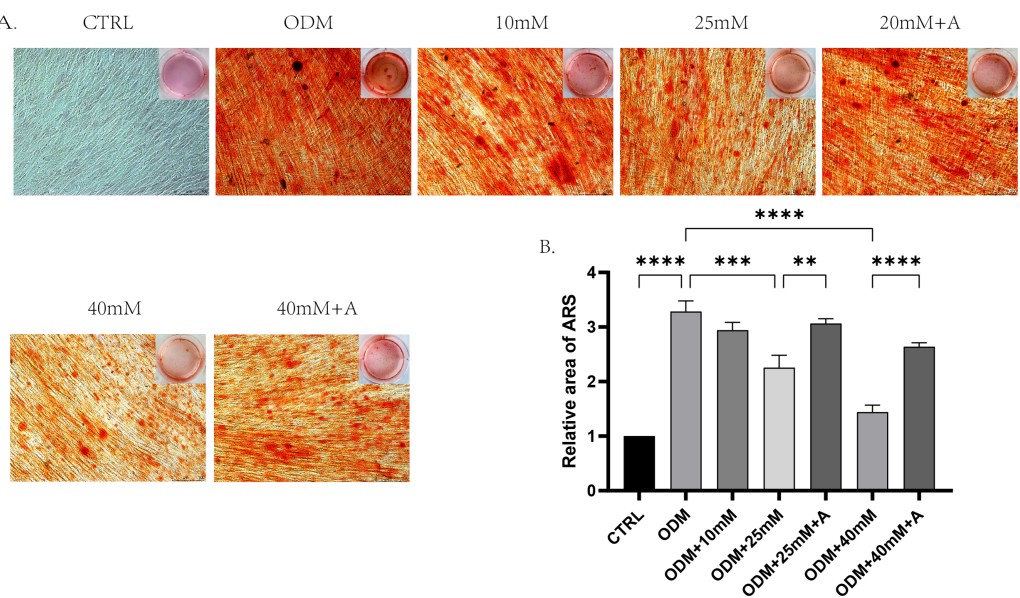

**Figure 5** **The effect of high glucose and ANXA2 on osteoblastic differentiation of PDLCs.** (A) Alizarin Red S staining at day 21 is shown, (B) Semi-quantitative analysis of ARS staining was performed by an Image J Analyzer. All data are presented as the means ± SEM from three independent experiments. $^{*}p < 0.05$, $^{**}p < 0.01$, $^{***}p < 0.001$. All data are presented as the means ± SEM from three independent experiments. $^{*}p < 0.05$, $^{**}p < 0.01$, $^{***}p < 0.001$.

for 4 h. Then, the protein expression of p62, LC3-II, were detected. The result did not provide statistical evidence of the changes in p62 in the high glucose condition. However, a significant increase in LC3-II/I ratio was observed after application of 1 μM rANXA2 compared to 40 mM(Fig. 8). Therefore, rANXA2 may enhanced the autophagy of PDLCs in high glucose condition.

# DISCUSSION

There is increasing evidence that PDLCs play a key role in maintaining the homeostasis of the periodontal microenvironment (*Yuan et al., 2023*). Diabetes mellitus, as a metabolic disease, is characterized by high levels of blood glucose, which can exacerbate periodontal destruction in patients with periodontitis. A large number of studies have demonstrated that the activity of periodontal ligament cells are inhibited in a high-glucose situation (*Chang et al., 2013*; *Li et al., 2023*; *Yu et al., 2024*). Therefore, it is imperative to find the mechanism underlying how high glucose damages periodontal ligament cells.

This study reveals that high glucose affects ANXA2 expression in PDLCs, and in addition, the administration of rANXA2 enhances autophagy, reduces ROS and delays cellular senescence, and promotes osteogenic capacity under high glucose conditions. Thus, ANXA2 attenuated the deleterious effects of high glucose on PDLCs, which may be involved in the relationship between diabetes mellitus and periodontitis.

In this experiment, we cultured PDLCs in medium containing different concentrations of glucose and found that their ANXA2 expression was decreased, a result that is consistent

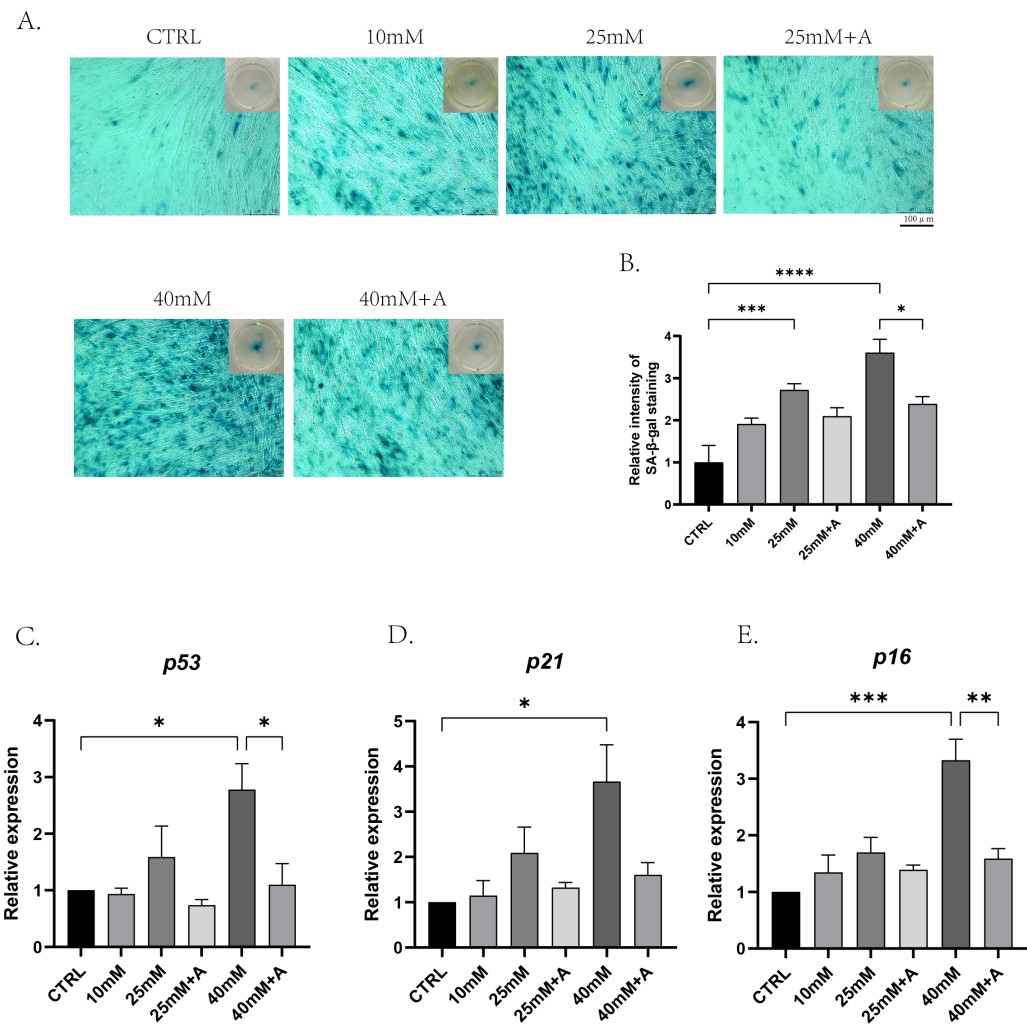

**Figure 6** **The effect of high glucose and ANXA2 on cellular senescence of PDLCs in medium with high glucose.** The PDLCs were cultured in medium under normal and high-glucose conditions with or without rANXA2. (A) The SA-β-gal staining at day 3. (B) Semi-quantitative analysis of SA-β-gal staining was performed by an Image J Analyzer. (C–E) The mRNAs of p53, p21, p16 were evaluated by qRT-PCR at day 3. All data are presented as the means ± SEM from three independent experiments. $^*p < 0.05$, $^{**}p < 0.01$, $^{***}p < 0.001$.

with other study (*Klabklai et al., 2022*). Also, our results showed that osteogenesis-related gene expression and matrix mineralization were decreased in PDLCs under high glucose conditions, which is supported by other study (*Zheng et al., 2019*). Interestingly, this result was inhibited by the addition of rANXA2. This suggests that ANXA2 is involved in the osteogenic mineralization process of PDLCs under high glucose conditions.

It is reported that high glucose induces oxidative stress in cells as well as cellular senescence which in turn causes numerous dysfunctions (*Tan et al., 2021*), which is consistent with our findings. Interestingly, our study showed that the rANXA2-added group reduced galactosidase activity and expression of senescence-related genes, as well as ROS,

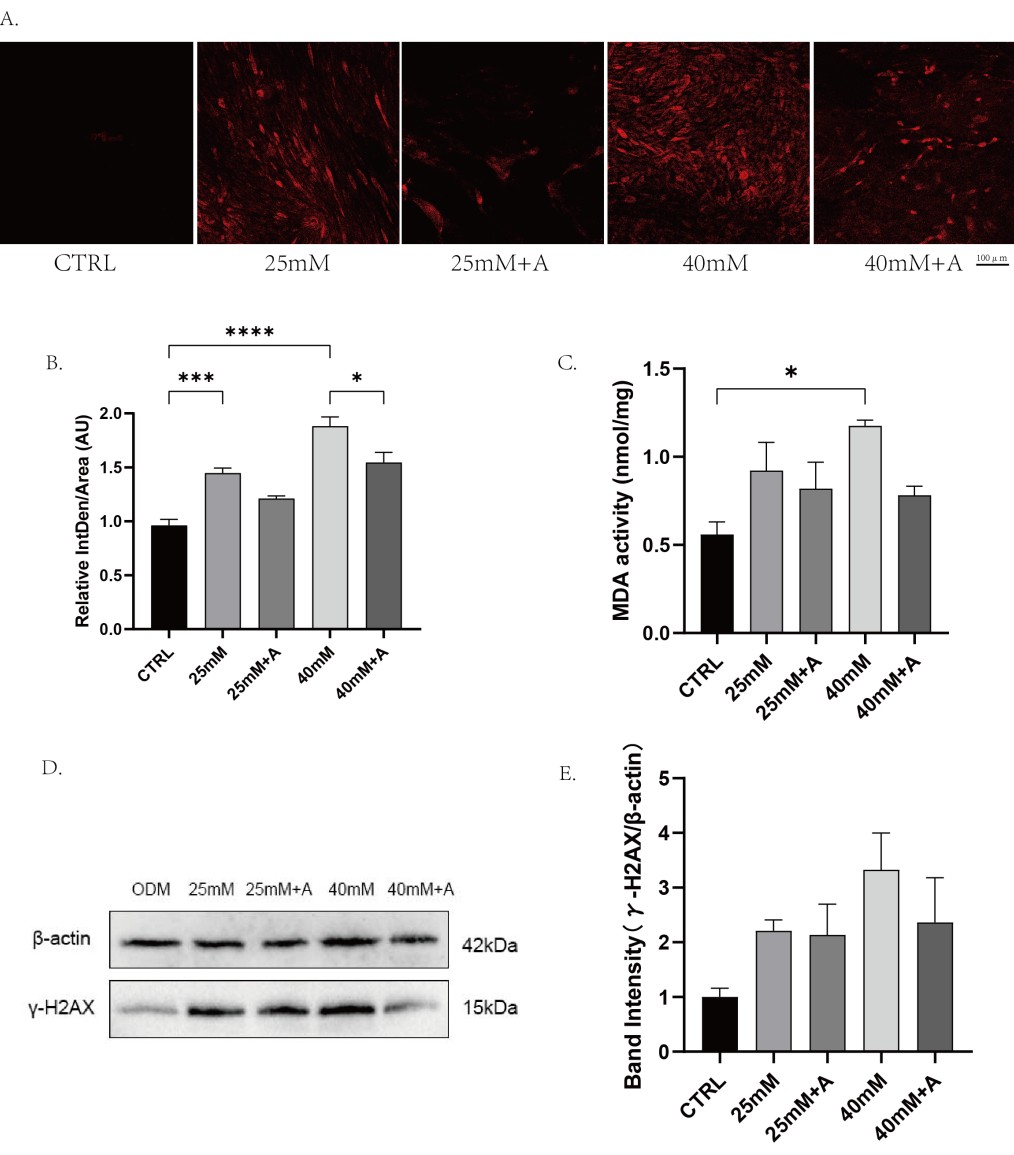

**Figure 7** **The effect of high glucose on oxidative stress of PDLCs in ODM.** (A) The ROS accumulation of PDLCs cultured in 25 mM and 40 mM, with or without rANXA2. (B) Semi-quantitative analysis of ROS accumulation. (C) The MDA activity of PDLCs in high glucose. (D) The expression of γ-HXA2 at protein level in PDLCs treated with HG. (E) Semi-quantitative analysis of γ-HXA2. All data are presented as the means ± SEM from three independent experiments. $*p < 0.05$, $**p < 0.01$, $***p < 0.001$.

which represent the level of oxidative stress, compared to the control group. Unfortunately, the above results on MDA and γ-H2AX were not obtained statistically, probably because the number of replications was not large enough, and more experiments are still needed to prove the above results. A large number of studies have shown that oxidative stress and cellular senescence can attenuate the osteogenic differentiation potential of PDLCs (*Jia et al., 2020*; *Kuang et al., 2020*; *Yang et al., 2021*), so we hypothesized that ANXA2 could reduce the level of oxidative stress and delay cellular senescence of PDLCs and thus promote
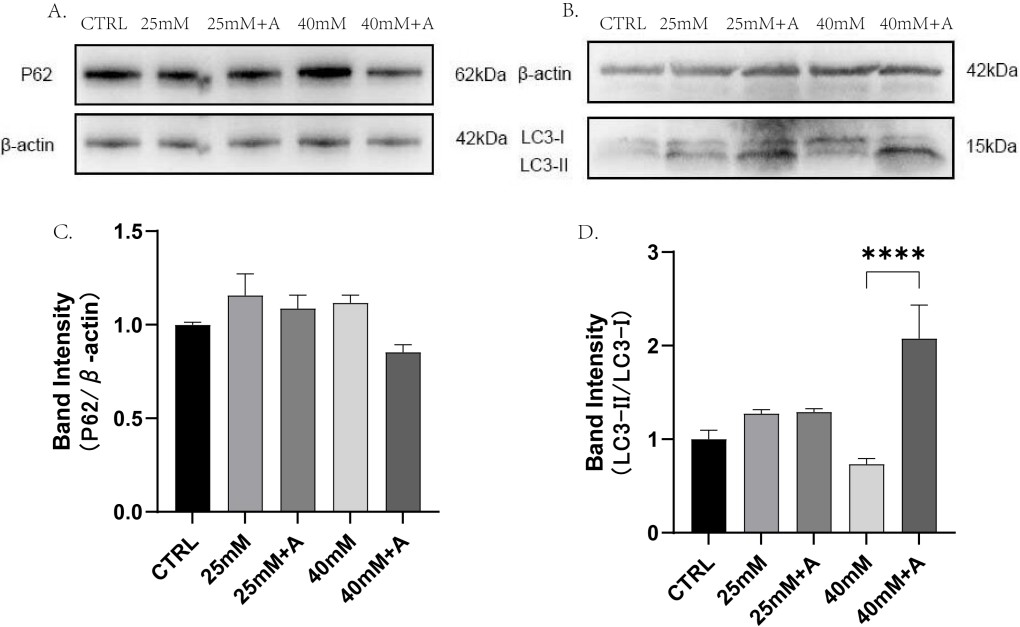

**Figure 8  The effect of high glucose and ANXA2 on autophagy of PDLCs.** (A) The expression of p62 at protein level in PDLCs. (B) The expression of LC3-II and LC3-I at protein level in PDLCs. (C) Semi-quantitative analysis of p62. (D) Semi-quantitative analysis of LC3-II/LC3-I. All data are presented as the means $\pm$ SEM from three independent experiments. $*p < 0.05$, $**p < 0.01$, $***p < 0.001$.

its osteogenic differentiation potential. This potential mechanism may be related to the activation of cellular autophagy.

Macroautophagy (hereinafter referred as autophagy) is a lysosome-dependent degradation process by which intracellular substances (long-lived proteins, damaged organelles) can be recycled (*Liu et al., 2022*) and it is critical in maintaining intracellular homeostasis (*Vargas et al., 2023*). Moderate autophagy favors cell survival in unfavorable environments, while excessive autophagy leads to cell death. It is closely related to the development of diseases (*Dikic & Elazar, 2018*). It had been shown that glycosylation end products (Argpyrimidine) bonded to RAGE inhibited autophagy and caused periodontal tissue destruction (*Li et al., 2022*). Yet, it has been suggested that autophagy inhibits osteogenic differentiation of periodontal stem cells (*Chang, Hsu & Wu, 2015*). In addition, autophagy dysfunction may cause mitochondrial dysfunction, which results in the production of large amounts of ROS (*Apostolova et al., 2023*), especially mitophagy.

In our experiments, the LC3II/I ratio was decreased in the high glucose condition compared to the control group, indicating that the autophagy level of the cells was inhibited, which is consistent with the results of other experiments (*Zhang et al., 2019*). However, there is data suggesting that high glucose can promote cellular autophagy levels (*Mecchia et al., 2022*), which may be related to the concentration of glucose applied and the duration. During the initial period of high glucose exposure, the increase in autophagy level may be a physiological response of cellular self-protection, and with the prolongation of time and the increase of glucose concentration, the cellular homeostasis is disrupted

and the autophagy level decreases and causes dysfunction. In our study, the addition of rANXA2 inhibited the high glucose-induced decrease in cellular autophagy levels. Therefore, we speculate that ANXA2 is involved in the regulation of cellular autophagy to reduce oxidative stress and cellular senescence, maintain cellular homeostasis, and thus promote osteogenic differentiation and matrix mineralization in PDLCs under high glucose conditions. Whether this is the situation and its underlying mechanisms will be our next step.

The current study focuses on *in vitro* experiments without the use of animal models. A large number of studies are still needed to further validate the relevant results. Moreover, this study has not yet addressed the mechanistic studies of the relevant signaling pathways and explained the underlying involvement of ANXA2 in the suppression of the osteogenic capacity of PDLCs in the high-glucose condition. We are planning to address these issues through further *in vitro* and *in vivo* studies.

## SUMMARIZATION

In conclusion, we demonstrated that ANXA2 expression was downregulated in PDLCs under high glucose conditions, and the addition of rANXA2 enhanced the autophagy level, reduced ROS and delayed cellular senescence, thereby promoting their osteogenic capacity. This paper suggests for the first time that ANXA2 may be involved in the development of severe periodontitis in diabetic patients, providing new ideas for relationship between periodontitis and diabetes.

### Funding
This study was supported by the Affiliated Hospital of Hangzhou Normal University, Zhejiang, China (PYJH202303). The funders had no role in study design, data collection and analysis, decision to publish, or preparation of the manuscript.

### Grant Disclosures
The following grant information was disclosed by the authors:
Affiliated Hospital of Hangzhou Normal University, Zhejiang, China: PYJH202303.

### Competing Interests
The authors declare there are no competing interests.

### Author Contributions
- Yanlin Huang conceived and designed the experiments, performed the experiments, authored or reviewed drafts of the article, and approved the final draft.
- Jiaye Wang performed the experiments, prepared figures and/or tables, and approved the final draft.
- Chunhui Jiang analyzed the data, prepared figures and/or tables, and approved the final draft.

- Minghe Zheng analyzed the data, prepared figures and/or tables, and approved the final draft.
- Mingfang Han analyzed the data, prepared figures and/or tables, and approved the final draft.
- Qian Fang analyzed the data, prepared figures and/or tables, and approved the final draft.
- Yizhao Liu analyzed the data, prepared figures and/or tables, and approved the final draft.
- Ru Li analyzed the data, prepared figures and/or tables, and approved the final draft.
- Liangjun Zhong analyzed the data, prepared figures and/or tables, and approved the final draft.
- Zehui Li conceived and designed the experiments, prepared figures and/or tables, authored or reviewed drafts of the article, and approved the final draft.

## Human Ethics

The following information was supplied relating to ethical approvals (i.e., approving body and any reference numbers):

The study was approved by the Ethics Committee of the Affiliated Hospital of Hangzhou Normal University (2023(E2)-KS-034).

## Data Availability

The raw data in the experiment are available in the Supplementary Files.

## Supplemental Information

Supplemental information for this article can be found online at http://dx.doi.org/10.7717/peerj.18064#supplemental-information.

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
