# Peer review of "ANXA2 promotes osteogenic differentiation and inhibits cellular senescence of periodontal ligament cells (PDLCs) in high glucose conditions"

_PeerJ, doi:10.7717/peerj.18064_

## Round 0.1 · original submission · Major Revisions

As you can see, all reviewers raised a number of critical concerns. Although two of the three reviewers recommended rejection, I decided to give you an opportunity to revise your manuscript in line with the indicated concerns. Please also provide a detailed response to the issues pointed by the reviewers.

Reviewer 1 ·

Basic reporting

Minor adjustments are necessary regarding the absence of periods in some sentences throughout the manuscript. Please, double-check.

The aims and hypothesis could be optimized by including information about verifying the effects of ANXA2 on reversing harmful effects of glucose on PDLC.

Experimental design

Even though the proposal is interesting and the authors conducted a series of experiments, the justification is not solid in the introduction and the design is confusing. The authors did not indicate any link between diabetic conditions and ANXA2 in the introduction, so why is this important to test?

Also, the authors are going back and forth with different concentrations of glucose in the experiments. For instance, some analyses tested 2s5 mM and 40 mM of glucose, others included 8 and 10 mM, some tested only 40 mM, and the initial CCK8 analysis also included 55 and 70 mM. Moreover, the CCK8 for the effect of ANXA2 on recovering cell viability only tested 40 mM of glucose. There is no clear explanation for that.

Validity of the findings

Due to the absence of clarity about the concentrations of glucose and ANXA2, the results section should be fully edited accordingly for consistency. Justification on the tested concentrations should be added to the manuscript.

Also, the authors tested the efficacy of ANXA2 on the medium with high glucose and not exactly on the cells previously exposed to high glucose. In that case, ANXA2 only reduces the harmful effects of glucose and does not reverse the conditions of cells previously affected by the stress of glucose exposure. The cells should be first treated with glucose for a certain period of time before the treatment with ANXA2 to simulate a diabetic environment.

The authors claim in the second paragraph of the discussion that ANXA2 is expected to be a promising therapeutic target for diabetes and periodontitis. It is hard to make that claim based on the experiments performed in this study.

The discussion section should include the limitations of the study and future methods to validate the claims, such as in vivo experimentation in diabetic rodents.

After rearranging the manuscript according to the performed experimental design, the conclusion should also be reorganized to match the findings.

Additional comments

Concentrations of glucose and ANXA2 should be first described in the methods.
The authors should report the absorbance of the readings for the ALP and MDA.
Describe in detail the SA-β-gal protocol.
The number of cells cultured and time points for the RT-PCR should also be reported in the methods.
Why the expression of ALP was not investigated by PCR as well?
All the figures indicate the concentration of glucose in G instead of mM. Please adjust that.
Figure 4F is missing ALP data for the control group.
It is not clear why the authors did not test 50 mM and 70 mM of glucose for all the experiments if they were the ones with the most harmful effects on the PDLCs.

Reviewer 2 ·

Basic reporting

The direct relationship between periodontitis and ANXA2 in the absence of hyperglycemia is unclear, as is the connection between ANXA2 and diabetes. Therefore, the text does not adequately justify the need for this scientific study. Additionally, some parts of the text lack necessary references, such as in the first paragraph of the introduction (lines 40, 43-45). In line 44, the authors use the term "DIABETIC PERIODONTITIS," which is not part of the current classification of periodontal diseases published in 2018. The relationship between periodontitis and diabetes has been studied since the 1960s. Still, the authors present it as though it were a recent or understudied research area when they use the word "now". The introduction consists of disconnected paragraphs that fail to adequately demonstrate how the study fits within the existing body of knowledge. Furthermore, the authors occasionally reference "various studies" but cite only a single study.

Experimental design

In general, the number of cells used in some of the experiments (such as qPCR) is lacking. The description of glucose concentrations is very confusing. The figure describes glucose concentrations in G instead of mM.

Validity of the findings

The authors addressed the objectives of the study; however, they overstate the implications by suggesting that their findings could lead to a new target for periodontal treatment. The text suggests that the study's results could have significant implications for periodontitis treatment development; however, these conclusions appear premature and are not adequately supported by the data presented. Since this is the first study indicating that ANXA2 might be involved in the development of periodontitis, discussing treatment strategies at this stage is premature. Future expectations should focus on subsequent research steps rather than "the ultimate goal" of the research line. If the authors believe in this potential, the text should be rewritten and supported with robust references to justify this association. Currently, none of the cited references relate to periodontal treatment. The article lacks a critical analysis of the results and a balanced discussion of their limitations and potential alternative interpretations.

Reviewer 3 ·

Basic reporting

Overall Feedback:
Thank you for the opportunity to review "ANXA2 promotes osteogenic differentiation and inhibits cellular senescence of periodontal ligament cells (PDLCs) in high glucose conditions." The article demonstrates the modulation of ANXA2 in high glucose environments concerning osteogenesis, autophagy, oxidative stress, and senescence, and it also explores the use of r-ANXA2 to mitigate these effects.
In my view, the article provides sufficient field background and context, the figures are well presented, and the results somewhat support the suggested hypothesis. However, I have suggestions to improve the reporting:

1. Abstract:
o I suggest highlighting the different glucose concentrations used in the methods and results sections in parentheses. Additionally, include the concentration used for subsequent experiments, as well as the concentration of r-ANXA2. For the main results, also highlight the p-values found in parentheses.

2. Introduction:
o I recommend exploring the role of PDLCs in periodontal regeneration in more detail. The authors only mentioned that it is well-documented but did not describe its significant role.

3. English Language Review:
o I suggest a more thorough review of the English language. Although it is well-written, some expressions are confusing. For example, on line 60, the authors write "Someone found," and on line 90, they write "In a word." These terms are not commonly used.

By implementing these suggestions, the clarity and impact of the article could be significantly enhanced.

Experimental design

Here, I have suggestions and some questions that I would appreciate being clarified:

4. Methods:

4. 1. Presentation:
o I suggest aligning the order of the methods with the results presentation. For instance, the authors presented "2.9 Western Blot Analysis" (line 141), but the Western Blot data were presented in "3.2 Results" (line 164). Following the same sequence in methods and results, when possible, will help the reader understand the authors' rationale and more easily interpret the results. Additionally, I recommend stating the objective of each method before introducing it. E.g: To assess the effect of glucose on the expression of the ANXA2 protein...

4. 2. Isolation of Populations:
o Were the patients systematically healthy? Please clarify this in section 2.1. Another point is the age of the patients. The authors mention that PDLCs were extracted from patients aged 10-25 years. I'm curious because third molars usually do not erupt in patients at the age of 10. Were these teeth in a developmental stage? Please clarify.

4. 3. Cell Viability Test:
o Please include the concentrations of glucose and r-ANXA2 used in this section. Although we can see this in the figure, it is important to document it in the methods. Additionally, were these concentrations based on any previous studies? How did the authors choose the glucose and r-ANXA2 concentrations? How were glucose and r-ANXA2 solubilized? Directly in the culture medium?

4. 4. Presentation of Groups:
o After the cell viability section (Section 2.2), I suggest the authors clearly present the working groups with their respective concentrations. This can be done by creating a new section (e.g., "Analyzed Groups" or another suitable title) and describing each group. This would greatly facilitate the reader's understanding. For the AR-S and qRT-PCR data, do not forget to mention in the respective sections (2.4 and 2.8) that the groups were in osteogenic medium and that the ODM group was also included.

4.5 Cell Concentrations:
o My main question is regarding the cell concentrations used in the study. For different analyses, using the same 6-well culture plates, the authors used different cell concentrations: 1x10^5 for ALP activity, 5x10^5 for AR-S, and 1x10^6 for intracellular ROS in 24-well plates (higher then 6-well plates). Wouldn't using different concentrations impact the results? Please explain the reasoning behind these choices.

4.6 Quantification of ROS:
o In section 2.6, the authors mention that five images were obtained by confocal microscopy using random selection. Please explain in more detail how this selection was performed.

4.7 Statistical Analysis:
o Did the authors use any analysis to check the data distribution? Were the data normal or non-normal? Was any normalization performed?
o Regarding the statistical tests, did the authors consider using two-way ANOVA to evaluate differences concerning the days when possible? If not, please explain the rationale for using only one-way ANOVA.

4. 8 Minor Points:
Some important information is missing in the sections:
o Indicate the source of glucose used during the study in section 2.2.
o Indicate the osteogenic medium used in section 2.4.
o Specify which confocal microscope was used in section 2.6.
o Indicate if the MDA test was conducted according to the kit instructions in section 2.7.
o Describe how total RNA was quantified in section 2.8.
o Specify the days (period) of RNA and protein collection for cells in sections 2.8 and 2.9.
o Specify the primary antibody used in section 2.9.
o Specify the primary antibody used in section 2.9.

I suggest reviewing the materials and methods section to clarify these details, ensuring that the experiments are clear and reproducible.

Validity of the findings

5. Results
Regarding the results, I believe they are well established for the most part and well presented. Please consider the suggested modification in the materials section concerning the order; I suggest maintaining the order of the presented results and modifying the order of the methods.
However, I have some observations:

5. 1. Viability:
o The authors describe the effect as dose-dependent, which it was not. At lower concentrations, there was no difference compared to the control, and the difference between the groups was not demonstrated. Please review this statement.

5. 2. Osteogenesis:
o There is sufficient data to assert that the low concentration of ANXA2 in the presence of glucose can indeed be one of the causes of reduced differentiation (Section 3.4), as differentiation is enhanced with the administration of r-ANXA2. However, regarding the relationship between these data and senescence and oxidative stress (Sections 3.5 and 3.6, respectively), there is insufficient data to claim that the results of senescence and oxidative stress impact osteogenesis, as the authors did not present data in osteogenic condiction (ODM). If the CTRL group refers to the basal medium (cells in standard culture medium), then the effect on osteogenesis cannot be evaluated. Unless the authors have, or can conduct, a new experiment in an osteogenic medium, I suggest reviewing the statements in lines 208 and 220, as well as the discussion and conclusion. I am open to discussion if the authors think differently and can substantiate their observations.

4. 3. Minor Points:
o Please refrain from making assertions about results if no statistical differences were found. If no significant differences were observed, it indicates that the groups are statistically equal. For instance, review the statements in lines 217 and 226 accordingly, and describe the results based on the statistical findings.
o Regarding the presentation of figures, some sections do not reference the figures (e.g., Section 3.2). Please review each result and introduce the corresponding figures in parentheses. Avoid directly referencing the figure (line 175).I suggest consistently following the same pattern for presenting figures in the text.

5. Discussion
o Regarding the discussion, I believe it is well-founded and I have no comments except for the need to review the English and the relationship of the osteogenesis data with the other findings, as previously mentioned.

6. Figures:
o Fig4. Considering the mineralization experiments and gene expression analysis of osteogenic genes by qRT-PCR were performed in osteogenic medium, I suggest modifying the groups on the X-axis to include: CTRL, ODM, ODM +10G, ODM +25G, ODM +25+A, ODM +40G, and ODM +40G+A.

Additional comments

Overall, the article presents important and interesting data. In the context of periodontal regeneration, and understanding the intricate relationship between diabetes, periodontitis, and the restoration of periodontal tissues, I believe this work provides valuable results and insights into a relatively unexplored marker (ANXA2). However, the article requires the suggested adjustments to improve readability and to clarify the questions posed by this reviewer, addressing important methodological concerns and some results.

---

## Round 0.2 · Minor Revisions

Please address the remaining concerns of the reviewer and amend manuscript accordingly

Reviewer 3 ·

Basic reporting

I am very pleased to have the opportunity to review this manuscript once again and to observe an improvement in the work. Most of the issues and suggestions I previously pointed out have been addressed and responded to with solid justification.

However, I still have a few minor comments for the authors to consider:

ABSTRACT:

1. I suggest specifying the exact p-value of the statistical result, rather than simply stating p<0.05.

Experimental design

METHODS:

1. In sections "2.3 Presentation of experimental groups and drug concentrations" and "2.4 Western Blot Analysis", the cell concentrations and type of culture plates should be described, as the authors have done in other sections. For example, in section 2.5 Alkaline Phosphatase (ALP) activity analysis:
“To examine the degree of PDLCs osteogenic differentiation, the ALP activity assay was performed when cultured in 6-well plates (1×10^5 per well) containing…”

2. In the "2.11 Statistical Analysis" section, please include the normality test used. This was well addressed by the authors in response to my previous inquiry, but I believe it is important to also include it in the manuscript.

Validity of the findings

Regarding the Results and Discussion sections, the organization has significantly improved. I have only a few suggestions:

1. I recommend avoiding the use of the word 'failed,' as seen in lines 267, 276, and 330. Consider rephrasing these sentences. For example: 'However, the current results did not provide statistical evidence that rANXA2 can inhibit this process.'

2. In lines 291 and 293, the text is somewhat repetitive. I suggest rephrasing the expression 'consistent with other studies' in line 293."

Additional comments

I hope the authors can consider the new comments to further enhance the quality of the article.

---

## Round 0.3 · accepted · Accept

Since all issues pointed by the reviewer were completely addressed, the revised manuscript is acceptable now.